# Compositional VQ Sampling for Efficient and Accurate Conditional Image Generation

## Abstract

Compositional diffusion and energy-based models have driven progress in controllable image generation, however the challenge of composing discrete generative models has remained open, holding the potential for improvements in efficiency, interpretability and generation quality. To this end, we propose a framework for controllable conditional generation of images. We formulate a process for composing discrete generation processes, enabling generation with an arbitrary number of input conditions without the need for any specialised training objective. We adapt this result for parallel token prediction with masked generative transformers, enabling accurate and efficient conditional sampling from the discrete latent space of VQ models. In particular, our method attains an average error rate of $19.3\%$ across nine experiments spanning three datasets (between one and three input conditions for each dataset), representing an average $63.4\%$ reduction in error rate relative to the previous state-of-the-art. Our method also outperforms the next-best approach (ranked by error rate) in terms of FID in seven out of nine settings, with an average FID of $24.23$, and average improvement of $-9.58$. Furthermore, our method offers a $2.3\times$ to $12\times$ speedup over comparable methods. We find that our method can generalise to combinations of input conditions that lie outside the training data (e.g. more objects per image for Positional CLEVR) in addition to offering an interpretable dimension of controllability via concept weighting. Outside of the rigorous quantitative settings, we further demonstrate that our approach can be readily applied to an open pre-trained discrete text-to-image model, demonstrating fine-grained control of text-to-image generation. The accuracy and efficiency of our framework across diverse conditional image generation settings reinforces its theoretical foundations, while opening up practical avenues for future work in controllable and composable image generation.

## 1 Introduction

Compositional generalisation in deep learning is the capacity of a trained model to respond correctly to *unfamiliar combinations* of *familiar concepts* (1). The ability of deep learning models to perform compositional generalisation is considered to be a pre-requisite for human-like artificial general intelligence (2), since compositional generalisation is something humans do remarkably well with relatively few training examples (3). Successful compositional generalisation remains an ongoing challenge in conditional image generation (4), which involves sampling from very high-dimensional spaces and is the focus of this work.

In the area of diffusion-based and energy-based image generation approaches, earlier works (5; 6; 7) have proposed methods for improving controllability of image generation via *composition* of energy-based and diffusion models. This family of methods enables *conjunction* and *negation* of input concepts by composing the probabilistic outputs of several feed-forward operations, each supplied with a different input condition. While composed techniques exceed non-composed baselines in terms of accuracy and image quality (7), such approaches do not extend to image generation models with *discrete* sample spaces, which offer a number of trade-offs and outright improvements over their continuous counterparts (8; 9; 10; 11).

Discrete image generation methods include autoregressive sampling approaches (9), and more recently discrete absorbing diffusion (10; 12; 11; 13) (also referred to as "masked generative transformers")

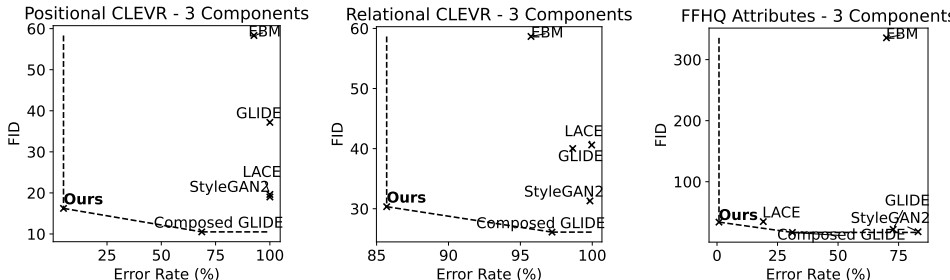

Figure 1: Overview of our approach. Generation begins with a fully masked discrete representation of an image. At each generation step, *unconditional* and *conditional* unmasking probabilities are obtained, conditioned on the unmasked state and input attributes. Next, our discrete compositional framework is applied, before sampling from the resulting distribution and unmasking a random selection of tokens. This is repeated until a fully unmasked representation of an image is obtained, which is finally decoded into an image.

Figure 2: Scatter plots of compositional generation error vs FID on 3 datasets (3 input components). Our method lies on the Pareto front of all results (see Appendix C for full scatter plots) while achieving lowest or joint lowest error among the baselines.

which enables a dramatic speed-up via the parallel prediction of discrete latent codes. These methods have been proposed as discrete alternatives to continuous image generation methods (EBM (14) and continuous diffusion (15)). Discrete methods for image generation are coupled to some method of mapping between high-dimensional pixel space and the discrete latent space, most commonly VQ-VAE (8) and VQ-GAN (9). Parallel prediction methods in particular (10) have a number of advantages over their continuous analogues, most notably in generation speed, in addition to image quality and diversity (10; 11). Until now, these advantages have remained mutually exclusive with the advantages of compositional generation, which have thus far been limited to *continuous* approaches only(6; 7).

To address this limitation, we propose a robust compositional image generation approach that brings the efficiency advantages of composable generation to the realm of discrete generative approaches. We derive specific formulae to represent *conjunction* and *negation* operations on the logit outputs of discrete conditional generative models, inspired by the tried-and-tested product-of-experts paradigm (16). Our method offers theoretically-principled and fine-grained control of generated outputs, while maintaining the significant speed advantages of parallel token prediction ($2.3\times$ to $12\times$ speedup on our hardware, Section 4). As an added benefit, our method attains state-of-the art error rate on 9 out of 9 quantitative settings (3 datasets with 1, 2 or 3 input components), equating to an **average reduction in error rate of 63.4%**. Meanwhile, our method outperforms the next-best approach (ranked by error rate) in terms of FID in 7 out of 9 settings. These results demonstrate the utility of our compositional approach as a viable or even preferable alternative to existing conditional image generation methods.

We further show that our method can be applied to an open pre-trained text-to-image parallel token model (aMUSEd (13)) with outstanding visual results (Figure 3). The successful application of our method to an out-of-the-box and open source pre-trained model broadens the potential for the use of our approach in practical controllable image generation applications. We discuss the broader societal impact of our work in Section 5. The effective implementation of our framework across

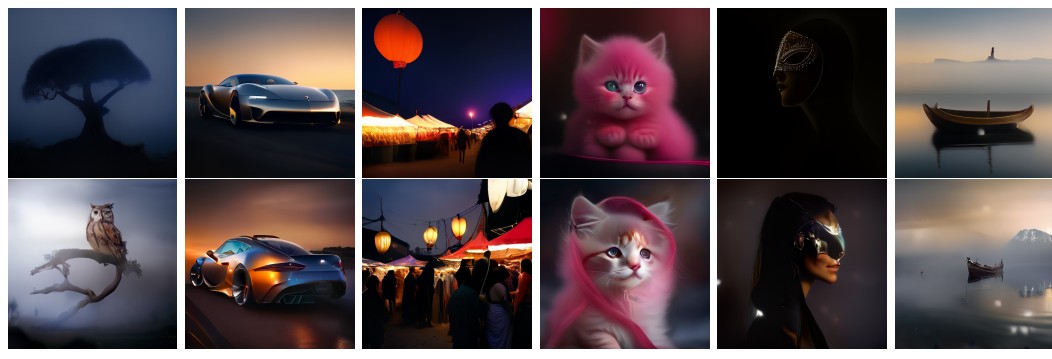

Figure 3: Compositional text-to-image results with captions (zooming recommended). *Top:* single-prompt baseline. *Bottom:* composed multi-prompt (ours). Our framework allows for the composition of multiple conditions, conferring an advantage over the single-prompt baseline.

diverse conditional image generation scenarios offers empirical validation for the mathematical underpinnings of our contribution, meanwhile laying the foundations for prospective practical applications in controllable conditional image generation.

## 2 RELATED WORK

**Product of experts:** The concept of compositional generalisation of discrete outputs relates directly to earlier approaches to classification, in particular the "product of experts" (PoE) framework (16). PoE combines multiple probabilistic models operating on the same input space, leveraging each of their individual "expertise" to improve overall generalisation performance. This approach has been shown to be particularly powerful in contexts involving high-dimensional inputs (16), in which each "expert" focuses on a subset of constraints, with the final outputs assigning high probability to outputs that satisfy constraints imposed by *all* experts. Our work seeks to adapt this idea to the compositional generation of images in discrete spaces, and is the first (to our knowledge) to apply this idea to iterative generative processes over multiple successive time-steps.

**Continuous compositional models:** Earlier works proposed methods for conditioning energy-based (6; 5) and diffusion models (7) respectively based on the conjunction and negation of input attributes, drawing on formal analogues to PoE (16). (7) introduces an approach that enhances the capabilities of text-conditioned diffusion models in generating complex and photo-realistic images based on textual descriptions. Our method draws inspiration from these ideas idea (principally in proposing formal derivations of probabilistic conjunction and negation operators), however our mathematical formulation diverges significantly due to the fact that our method applies to *discrete* iterative approaches (compare with EBM and diffusion, which operate in a continuous output space (14; 15)). The novelty of our derivation, and its application to discrete generation go beyond the literature by offering a novel formulation for composing iterative discrete generation models, while empirically achieving state-of-the-art in image generation accuracy while attaining competitive FID scores. All compositional approaches to date, our own included, somewhat resemble the mathematical form of **classifier-free guidance (CFG)** (17), which is effective in obtaining extra dimension of controllability over the outputs of both continuous and discrete diffusion models (13).

**Composition in sequential tasks:** Earlier work in reinforcement learning (18) has sought to use ideas relevant to our own for composing *policies* for the purposes of compositional generalisation in multi-timestep environments. The main idea shared with our work is that of *multiplying* constituent "primitives" (as opposed to *additive* composition, as in mixture-of-experts (19)). According to

(18), the benefit of this multiplicative approach over mixture-of-experts is that multiple policies can be expressed in the same sequence. This shares some superficial similarities with our work on compositional discrete image generation, where multiple attributes are required to be expressed in the same output image, however it differs significantly in both the application and the formalism.

**Discrete representation learning and sampling:** Discrete representation learning has emerged as the discrete counterpart to continuous VAE approaches (8; 20). Discrete representation learning is based on the concept of vector-quantization (VQ) (21), whereby features from a continuous vector space are mapped to an element of a finite set of learned codebook vectors. This VQ family of models includes VQ-VAE (8) and VQ-GAN (9). VQ-family approaches require a secondary prior model to be trained to sample from the discrete latent space, which can be computationally expensive at both train- and inference- time (8; 9; 10). A more recent sampling approach aims to address this with parallel token prediction using a transformer encoder (10; 11; 13) (akin to masked language modelling (22)), which introduces a controllable trade-off between sample speed and generation quality, as well as the ability to control the diversity of outputs via temperature. The per-image generation time is still linear in the size ($W \times H$) of the image, albeit with a smaller constant than autoregressive models (10). The advantages of parallel token prediction over autoregressive approaches motivate our work in producing a method for composing discrete image sampling approaches.

## 3 METHOD

In this section we present our method for composing generative models for controllable sampling from discrete representation spaces of images. We first derive a novel formulation that directly informs our method of composing conditional distributions over discrete spaces. Next we show how this extends generally to all discrete sequential generation tasks, in which categorical variables are sampled iteratively to produce a complete sample (e.g. via autoregressive or masked models). We show how this result can be specifically adapted for conditional parallel token prediction (10; 11) to achieve high-quality and accurate controllable image synthesis. This is further enhanced by concept weighting, which allows the relative importance of input conditions to be increased, decreased or negated to the desired effect. We note that similar results can be obtained for other types of generative model (provided they are iterative and discrete, see Appendix D.3 for example). This section lays the groundwork for our later experiments with compositional sampling from the latent space of VQ-VAE and VQ-GAN.

### 3.1 COMPOSING CONDITIONAL CATEGORICAL DISTRIBUTIONS

Our framework is derived from the simplifying assumption that the input conditions $c_1, ..., c_n$ are independent of each other conditional on the output variable $x$, i.e. $P(c_1, c_2|x) \propto P(c_1|x)P(c_2|x)$ for any pair of distinct attributes $(c_1, c_2)$ (this is the product of experts assumption (16). A consequence of this is that probability of an outcome $x$ given two or more conditions $c_1, c_2, ...$ is proportional to the product of the probabilities of each condition given $x$. Intuitively, taking the product of different categorical distributions in this way is analogous to taking the intersection of the sample spaces of two or more conditional distributions, thus (in the ideal scenario) resulting in samples which embody all of the specified conditions (16). We discuss the benefits and limitations of this assumption in Section 5. Following previous work with composition of continuous models (2; 7), we factorize the distribution of a $k$-way categorical variable $x$ conditioned on $n$ variables as follows:

$$P(\boldsymbol{x}|\boldsymbol{c}_1, ..., \boldsymbol{c}_n) \propto P(\boldsymbol{x}) \prod_{i=1}^{n} P(\boldsymbol{c}_i|\boldsymbol{x}) \tag{1}$$

Applying Bayes' theorem (23), this can be re-written as:

$$\begin{aligned} P(\boldsymbol{x}|\boldsymbol{c}_1, ..., \boldsymbol{c}_n) &\propto P(\boldsymbol{x}) \prod_{i=1}^{n} \frac{P(\boldsymbol{x}|\boldsymbol{c}_i)P(\boldsymbol{c}_i)}{P(\boldsymbol{x})} \\ &\propto P(\boldsymbol{x}) \prod_{i=1}^{n} \frac{P(\boldsymbol{x}|\boldsymbol{c}_i)}{P(\boldsymbol{x})} \end{aligned} \tag{2}$$

We are able to eliminate the $P(\boldsymbol{c}_i)$ term in (2) as a consequence of normalising the values of $P(\boldsymbol{x} = x_i|...)$ for all $x_i$ such that they sum to 1 (see Appendix D.1 for full derivation).

### 3.2 COMPOSITION FOR SEQUENTIAL GENERATIVE TASKS

So far we have shown how our approach applies to conditional generation with a single categorical output $x$. In practice, many generative tasks involve sampling multiple categorical variables (tokens or latent codes) over a number of time steps (24; 25) where the sampling of a new state $s_{t+1}$ at each successive step $t$ is conditioned on the previous state (in addition to the specified conditions $c_1, ..., c_n$). Formulating this alongside the result in (2) gives the following general expression for composing discrete sequential generation tasks (see Appendix D.2 for further explanation):

$$P(s_{t+1}|s_t, c_1, ..., c_n) \propto P(s_{t+1}|s_t) \prod_{i=1}^{n} \frac{P(s_{t+1}|s_t, c_i)}{P(s_{t+1}|s_t)} \tag{3}$$

We observe that this applies generally to any generative process in which each successive step is conditioned on the output of previous steps, including autoregressive language modelling (25), masked language modelling (26), as well as autoregressive (9) and non-autoregressive (10) approaches for image generation. In the remainder of this paper, we maintain a particular focus on conditional parallel token prediction, which we use for composed sampling from the latent space of VQ-VAE (8) and VQ-GAN (9) for high-fidelity image synthesis.

A key practical consideration concerning the result in (3) is that estimates must be obtained for each conditional probability distribution $P(s_{t+1}|s_t, c_i)$ in addition to $P(s_{t+1}|s_t)$. In each of our experiments (Section 4) we ensure that, during training, conditional information is zero-masked with a set probability (0.1) per sample, thus allowing us to obtain $P(s_{t+1}|s_t)$ at inference time by supplying zeros in place of the condition encoding.

### 3.3 COMPOSED PARALLEL TOKEN PREDICTION

Parallel token prediction (10; 12; 11) is a non-autoregressive alternative to next-token prediction (9) for generative sampling from a discrete latent space. This allows a direct trade-off between sampling speed and image generation quality (10) by controlling the rate at which tokens are sampled. Parallel token prediction can be thought of as the gradual un-masking of a collection of discrete latent codes $z_0$ given the partial reconstruction from a previous time step (as well as additional conditioning information). This corresponds directly to the next-state prediction formulation in (3), but with the time labels $t$ reversed in order to reflect the "reverse process" which characterizes diffusion models (both continuous (27) and discrete (10)):

$$P(z_{t-1}|z_t, c_1, ..., c_n) \propto P(z_{t-1}|z_t) \prod_{i=1}^{n} \frac{P(z_{t-1}|z_t, c_i)}{P(z_{t-1}|z_t)} \tag{4}$$

In (4), $z_t$ is an intermediate, partially unmasked representation at each time step, and $z_{t-1}$ represents the distribution over image representations with fewer masked tokens. In practise, following earlier work with parallel token prediction, the model is trained to directly predict the fully unmasked representation $z_0$ (as opposed to intermediate states) so as to maximise training stability (10). At inference time, we compute the composed unmasking probabilities as:

$$P(z_0|z_t, c_1, ..., c_n) \propto P(z_0|z_t) \prod_{i=1}^{n} \frac{P(z_0|z_t, c_i)}{P(z_0|z_t)} \tag{5}$$

Where each $P(...)$ term corresponds to a feed-forward operation which takes a partially unmasked state as input, optionally with additional conditioning information $c_i$. Image representations can then be unmasked one or more tokens at a time, corresponding to a trade-off between sample speed (more tokens per iteration) and image generation quality (fewer tokens per iteration) (10).

### 3.4 CONCEPT WEIGHTING FOR IMPROVED CONTROLLABILITY

Following earlier work with composable diffusion models for image generation (7), we introduce an additional set of hyperparameters $w_1, ...w_n$ which correspond to the relative weight to be assigned

to each condition $c_1, ..., c_n$ respectively. Restating (3) in terms of log-probabilities and introducing these weighting terms gives:

$$\log P(s_{t+1}|s_t, c_1, ..., c_n) = \log P(s_{t+1}|s_t) + \sum_{i=1}^{n} w_i \left[ \log P(s_{t+1}|s_t, c_i) - \log P(s_{t+1}|s_t) \right] \quad (6)$$

This expression (6) can straightforwardly be manipulated into the appropriate form for parallel token prediction (4). A key observation here is that setting a concept weight $w_i$ to *negative* (e.g. $-1$) has the intuitive effect of *negating* the corresponding condition $c_i$ by excluding image representations which correspond to $c_i$ from the sample space. Altogether, the prompt-weighting approach provides an additional degree of controllability over model outputs by enabling conditions to be emphasized ($w_i > 1$), de-emphasized ($w_i < 1$) or even negated ($w_i < 0$) as desired. We demonstrate the practical utility of this feature in our qualitative experiments (Section 4). We do not include *disjunction* in our evaluation for reasons explained in Appendix D.4.

In practice, we use our compositional framework to sample from the discrete latent space of VQ-VAE (8) and VQ-GAN (9), which are powerful and practical approaches for encoding images and other high-dimensional modalities as collections of discrete latent codes (visual tokens) while producing high-fidelity reconstructions and generated samples.

### 3.5  DISCRETE ENCODING AND DECODING OF IMAGES

In order to compose categorical distributions for generating images, we must also define an invertible mapping between RGB images and discrete latent representations. We utilise a convolutional down-sampling and up-sampling (autoencoder) to map between RGB image space and latent embedding space. Following the original VQ-VAE (8) formulation, we employ nearest-neighbour vector quantization, in which encoder outputs are mapped to their nearest neighbour in a learned codebook. Specifically, for each encoder output vector $z_e$, the corresponding quantized vector is computed as the nearest codebook entry $e_c$, where

$$c = \arg\min_{j} ||z_e - e_j||_2 \quad (7)$$

and $e_0, e_1 ... e_{K-1}$ are entries in a learned vector codebook of length $K$.

Since the quantization step is non-differentiable, it is necessary to estimate the gradients during backpropagation. For this purpose, straight-through gradient estimation (28) is used, whereby during backpropagation the gradients are copied directly from the decoder input $z_c$ to the encoder output $z_e$. We use this vector quantization approach for all our experiments, which includes the embedding and commitment loss terms from the original VQ-VAE formulation (8).

## 4  EXPERIMENTS

### 4.1  DATASETS

Following earlier work (7) in evaluating compositional generalisation for image generation, we employ three datasets for training and evaluation: Positional CLEVR (29; 7), Relational CLEVR (29; 7) and FFHQ (30) (full description of datasets in Appendix A). These three datasets are chosen to represent a range of unique and challenging compositional tasks (conditioned on object position, object relations, and facial attributes respectively). For each of the three datasets, we train a VQ-VAE or VQ-GAN model to enable encoding and decoding between the image space and the discrete latent representation space, in addition to a conditional parallel token prediction model (encoder-only transformer) which learns to unmask discrete latent representations, optionally conditioned on an encoded input annotation.

### 4.2  MODEL TRAINING

We train a VQ-GAN model to reconstruct FFHQ samples at $256 \times 256$ resolution, as well as VQ-VAE models for each of CLEVR and Relational CLEVR at $128 \times 128$ resolution. These choices of

Table 1: Quantitative results (error rate and FID score) on the Positional CLEVR dataset

| Method | 1 Component | | 2 Components | | 3 Components | |
|---|---|---|---|---|---|---|
| | Err (%) ↓ | FID ↓ | Err (%) ↓ | FID ↓ | Err (%) ↓ | FID ↓ |
| StyleGAN2-ADA (33) | 62.72±1.37 | 57.41 | - | - | - | - |
| StyleGAN2 (34) | 98.96±0.29 | 51.37 | 99.96±0.04 | 23.29 | 100.00±0.00 | 19.01 |
| LACE (5) | 99.30±0.24 | 50.92 | 100.00±0.00 | 22.83 | 100.00±0.00 | 19.62 |
| GLIDE (35) | 99.14±0.26 | 61.68 | 99.94±0.06 | 38.26 | 100.00±0.00 | 37.18 |
| EBM (6) | 29.46±1.29 | 78.63 | 71.78±1.27 | 65.45 | 92.66±0.74 | 58.33 |
| Composed GLIDE (7) | 13.58±0.97 | 29.29 | 40.80±1.39 | 15.94 | 68.64±1.31 | **10.51** |
| Ours | **0.70**±0.24 | **13.76** | **1.82**±0.38 | **15.30** | **4.96**±0.61 | 16.23 |

Table 2: Quantitative results (error rate and FID score) on the Relational CLEVR dataset

| Method | 1 Component | | 2 Components | | 3 Components | |
|---|---|---|---|---|---|---|
| | Err (%) ↓ | FID ↓ | Err (%) ↓ | FID ↓ | Err (%) ↓ | FID ↓ |
| StyleGAN2-ADA (33) | 32.29±1.32 | 20.55 | - | - | - | - |
| StyleGAN2 (34) | 79.82±1.14 | 22.29 | 98.34±0.36 | 30.58 | 99.84±0.11 | 31.30 |
| LACE (5) | 98.90±0.30 | 40.54 | 99.90±0.09 | 40.61 | 99.96±0.04 | 40.60 |
| GLIDE (35) | 53.80±1.41 | **17.61** | 91.14±0.80 | **28.56** | 98.64±0.33 | 40.02 |
| EBM (6) | **21.86**±1.17 | 44.41 | 75.84±1.21 | 55.89 | 95.74±0.57 | 58.66 |
| Composed GLIDE (7) | 39.60±1.38 | 29.06 | 78.16±1.17 | 29.82 | 97.20±0.47 | **26.11** |
| Ours | **21.84**±1.17 | 30.00 | **56.94**±1.40 | 28.87 | **85.70**±0.99 | 30.34 |

resolution follow earlier work in compositional generation with these 3 datasets (7). We find in practice that VQ-VAE (without the adversarial loss) is sufficient for high-fidelity reconstruction of the two CLEVR datasets due to the smaller resolution and visual simplicity, while VQ-GAN is required for realistic reconstructions of FFHQ. Unlike (7), our choice of training regime produces FFHQ images directly at $256 \times 256$, so a post-upsampling step is not required during evaluation. We train with a perceptual loss (31) in addition to the MSE loss for all datasets (and the learned adversarial loss for FFHQ). Full details of model training are in Appendix F.

### 4.3 QUANTITATIVE EVALUATION OF COMPOSITIONAL GENERATION

For each dataset, following (7) we evaluate (compositionally) generated image samples according to both FID (Fréchet Inception Distance (32)) and binary classification accuracy (defined as whether a specified attribute, or collection of attributes, is present or not in the corresponding generated output image according to a pre-trained classifier). These two metrics are chosen in order to assess each model's ability to match the target distribution from the perspective of both perceptual image quality and visual accuracy. We conduct all quantitative experiments for 1, 2 and 3 attributes per image for all 3 datasets, totalling 9 quantitative experiments. We use a temperature of 0.9 when generating samples for our quantitative experiments. Details of how accuracy scores are obtained are in Appendix H.

In comparison to the 6 baseline results reported in (7), our method **exceeds or matches the accuracy of the previous state-of-the-art in all nine settings**, while attaining highly competitive FID scores across the three datasets. Particularly noteworthy are our accuracy results for Positional CLEVR, for which our method scores 99.30%, 98.18% and 95.04% on 1, 2 and 3 input components respectively, where the previous state-of-the-art scored 86.42%, 59.20% and 31.36% respectively (Table 1). We see similarly dramatic improvements much harder Relational CLEVR dataset (Appendix Table 2) and significant improvements on the FFHQ dataset (Appendix Table 3).

We speculate that the dramatic accuracy improvements offered by our method can be attributed to the fact that the introduction of the discrete representation learning step (VQ-VAE or VQ-GAN) facilitates the learning of an expressive and compositional "visual language" to represent images, while the conditional parallel token model offers a strongly regularised and highly calibrated model for the visual language. We conjecture that these effects compound to produce an efficient, accurate and robust compositional method.

Table 3: Quantitative results (error rate and FID score) on the FFHQ dataset

| Method | 1 Component | | 2 Components | | 3 Components | |
|---|---|---|---|---|---|---|
| | Err (%) ↓ | FID ↓ | Err (%) ↓ | FID ↓ | Err (%) ↓ | FID ↓ |
| StyleGAN2-ADA (33) | 8.94±0.81 | **10.75** | - | - | - | - |
| StyleGAN2 (34) | 41.10±1.39 | 18.04 | 69.32±1.30 | 18.06 | 83.04±1.06 | 18.06 |
| LACE (5) | 2.40±0.43 | 28.21 | 4.34±0.58 | 36.23 | 19.12±1.11 | 34.64 |
| GLIDE (35) | 1.34±0.33 | 20.30 | 51.32±1.41 | 22.69 | 72.76±1.26 | 21.98 |
| EBM (6) | 1.26±0.32 | 89.95 | 6.90±0.72 | 99.64 | 69.99±1.30 | 335.70 |
| Composed GLIDE (7) | 0.74±0.24 | 18.72 | 7.32±0.74 | **17.22** | 31.14±1.31 | **16.95** |
| Ours | **0.22±0.13** | 21.52 | **0.62±0.22** | 28.25 | **0.82±0.26** | 33.80 |

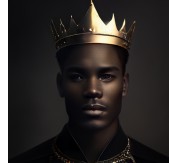 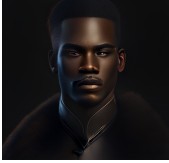 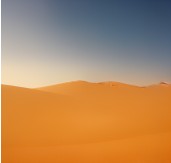 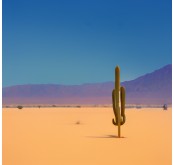 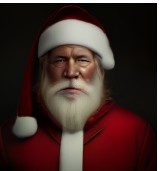 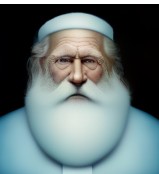

[*] "a king **not** wearing a crown, portrait" — "a king, portrait" **AND** (**NOT** "wearing crown") — [*] "a sunny desert with **no** sand landscape" — "a sunny desert, landscape" **AND** (**NOT** "sand dunes") — [*] "santa claus **not** wearing red, portrait" — "santa claus, portrait" **AND** (**NOT** "wearing red")

Figure 4: Concept negation with text-to-image (left baseline, right ours): Our method allows more precise control over the outputs of an existing pre-trained model (aMUSEd (13)). The baseline handles poorly the negation/removal of characteristics which are commonly co-occur with the subject of the image (e.g. a king with no crown). Each image is selected from three runs as the most representative of the prompt (ours and baseline).

## 4.4 QUALITATIVE EXPERIMENTS

Here we also qualitatively investigate the usefulness of our approach outside of the rigorous quantitative experimental settings. In particular, we investigate the controllability offered by logical conjunction and negation of prompts, as well as the qualitative effect of concept weighting. In addition to the models trained for our quantitative experiments, some of the experiments below apply our method to a pre-trained text-to-image parallel token prediction model ((11)). We choose the aMUSEd (13) implementation of MUSE because it is publicly accessible (as of the time of writing), in addition to being trained on a large, open dataset (36), which facilitates the open-ended generation which we aim to explore here. Additional qualitative results are in Appendix E.

**Concept negation:** Fig. 4 demonstrates the application of concept negation using aMUSEd text-to-image parallel token prediction (13). We compare each example against a single-prompt CFG baseline using the same model. We focus on problematic cases where the underlying text-image model is incapable of properly interpreting negation in the linguistic sense, which is especially pertinent when the concept being negated may be considered an essential characteristic of the concept from which it is being negated (e.g. "a king **not** wearing a crown"). Our method allows us to achieve more specific outputs without changing or fine-tuning the underlying model, even in cases where the underlying model fails to comprehend the original negated prompt (Fig. 4).

**Out-of-distribution generation:** In Fig.5 we demonstrate our model's ability to generalise to compositions of conditions that are not seen in training. We focus on the (positional) CLEVR dataset, in which individual training samples have at most 5 objects per image. Fig.5 contains generated samples for input conditions specifying between 6 and 8 objects per image. We make two key observations of Fig.5: (1) that our method successfully generalises outside the distribution of the training data and (2) that re-running the same input gives varied outputs, i.e. the model has not over-fit to always generate the same objects in the same position.

**Varying the concept weight:** Fig.6 illustrates the effect of varying the concept weighting parameter $w$ for a specific input condition (in this case, the weighting of the "no glasses" attribute of FFHQ). Keeping other concept weights the same ($w_{smile} = w_{male} = 3.0$), we vary $w_{no\_glasses}$ from $-3.0$ to $3.0$. The outputs in Fig.6 are consistent with the expectation that the concept weighting capability of our method should allow for an interpretable degree of controllability over the generated outputs.

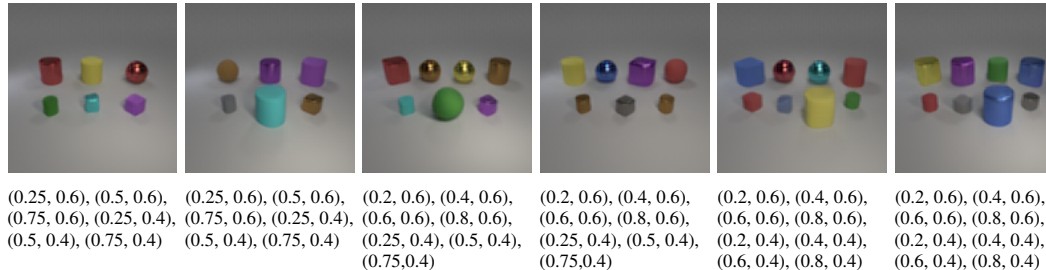

(0.25, 0.6), (0.5, 0.6), (0.75, 0.6), (0.25, 0.4), (0.5, 0.4), (0.75, 0.4)

(0.25, 0.6), (0.5, 0.6), (0.75, 0.6), (0.25, 0.4), (0.5, 0.4), (0.75, 0.4)

(0.2, 0.6), (0.4, 0.6), (0.6, 0.6), (0.8, 0.6), (0.25, 0.4), (0.5, 0.4), (0.75,0.4)

(0.2, 0.6), (0.4, 0.6), (0.6, 0.6), (0.8, 0.6), (0.25, 0.4), (0.5, 0.4), (0.75,0.4)

(0.2, 0.6), (0.4, 0.6), (0.6, 0.6), (0.8, 0.6), (0.2, 0.4), (0.4, 0.4), (0.6, 0.4), (0.8, 0.4)

(0.2, 0.6), (0.4, 0.6), (0.6, 0.6), (0.8, 0.6), (0.2, 0.4), (0.4, 0.4), (0.6, 0.4), (0.8, 0.4)

Figure 5: Compositional out-of-distribution generation: Positional CLEVR training images contain no more than 5 objects per image, but our compositional method allows 6 or more objects to appear in the same image via compositional sampling.

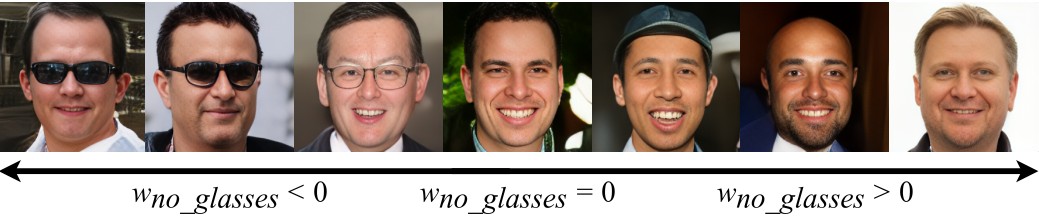

$w_{no\_glasses} < 0$        $w_{no\_glasses} = 0$        $w_{no\_glasses} > 0$

Figure 6: Effect of varying the $w_{no\_glasses}$ concept weight from $-3.0$ to $3.0$ while keeping $w_{male} = w_{smile} = 3.0$.

We observe that strengthening the negative weight increases the bias towards sunglasses, which we postulate is due to there being a smaller overlap between the distributions of sunglasses vs. no glasses, compared to the overlap between corrective glasses vs. no glasses. We emphasize that the lack of glasses for the $w_{no\_glasses} = 0$ case does not signify a failure, but rather that the model output is agnostic to the presence or lack of glasses. We include similar visualisations for the other two FFHQ attributes in Appendix E.2.

### 4.5 PARAMETER COUNT AND SAMPLING TIME

Table 4 contains a comparison of our method to the most similar methods in the literature (composed EBM (6) and composed GLIDE (7). We compare our method against these methods in particular for 2 reasons: (1) they are compositional and iterative like our own method, and (2) they are generally closest to ours (lowest) in terms of error rate on the three datasets studied. We compare in terms of total parameters and the time taken to generate both a single image and a batch of 25 images on our hardware (NVIDIA GeForce RTX 3090) with 3 input conditions (Positional CLEVR dataset). Runs of baseline methods use the official PyTorch implementations from (7) with default settings (corresponding to the baseline results in Tables 1,2 and 3). The results in Table 4 show that our method runs in a fraction of the time of existing approaches while having a comparable number of parameters (and smaller error rate: see Tables 1,2, 3)). Altogether, we see a 2.3× to 12.0× speedup across our speed experiments compared with the baselines.

## 5 DISCUSSION

Through varied quantitative and qualitative experiments, we have demonstrated that our formulation for compositional generation with iterative sampling methods is readily applicable to a range of tasks for both newly trained and out-of-the-box pre-trained models. We demonstrated state-of-the-art performance in terms of the error rate of the generated results, in addition to obtaining competitive sample quality as measured by FID scores. This is achieved with minimal extra cost in terms of memory, since only the log-probability outputs need to be retained at inference time. The simplicity of our method offers further advantages, including ease of implementation (facilitating integration with existing discrete generation pipelines) in addition to improved interpretability, since the composition operator can be thought of as directly taking the "intersection" between two discrete distributions.

Table 4: Parameter counts and sample times for 3 input components (ours vs. baselines)

| Method | Total parameters (millions) | Sample time / img (s) | Sample time / batch of 25 (s) |
|---|---|---|---|
| EBM | 33 | 5.99±0.17 | 108.57±0.93 |
| Composed GLIDE | 385 | 4.92±0.17 | 73.92±0.70 |
| Ours | 108 | **2.11**±0.39 | **9.08**±0.39 |

The strong quantitative metrics of our method are complemented by its *out-of-distribution* generation capability and *controllability*. The significance of the results of our quantitative experiments is further reinforced by the fact that we used the same experimental settings for all three of the datasets studied, without extensive fine-tuning of hyperparameters, training runs or model architecture. Our method further provides a $2.3\times$ to $12\times$ speedup over comparable approaches on our hardware.

### 5.1 LIMITATIONS

Similarly to compositional methods for continuous processes (7), our method requires $(n + 1)$ times the number of feed-forward operations compared to standard iterative approaches of the same architecture, where $n$ is the number of conditions imposed on the output. This is a direct consequence of the mathematical formulation of the approach, however this is largely mitigated by the fact that our method can produce accurate and high-qaulity outputs in only a small number of iterations (6; 7).

Our method makes a strong assumption that input conditions are independent ($P(c_1, c_2) = P(c_1)P(c_2)$ for all conditions $c_1, c_2$). It is possible in practical scenarios that this underlying assumption of our approach is in some way violated, for example due to biases in the training data. The importance-weighting capability of our method can mitigate this in part by allowing the user to compensate for potential biases, however we speculate that greater robustness would be better achieved through an unbiased backbone model. Training unbiased models for image generation is beyond the scope of this work and remains an open challenge, especially in the context of text-to-image generation (37). Further to this, we have not yet explored principled methods for choosing the condition weighting coefficients $w_i$, which may be an interesting direction for future work (e.g. producing a learned concept-weighting policy).

### 5.2 BROADER IMPACTS

We have shown that our method can be applied directly to a publicly available pre-trained discrete text-to-image model without any fine-tuning to achieve fine-grained control over visual generation. While this presents an opportunity for positive impact by enabling creative works, we also wish to raise attention to the broader impact of our work from the perspectives of both *societal bias* and *misuse*. Image generation techniques in particular can be susceptible to perpetuating or even amplifying societal biases (38), and our method will inherit whatever biases are present in the training data or pre-trained model. In addition, readily accessible and controllable image generation presents opportunities for *misuse*, for example for the purposes of misinformation (39) as well as defamation and impersonation (40).

## 6 CONCLUSION

We have proposed a novel method for enabling precisely controllable conditional image generation by composing discrete iterative generative processes. The empirical success of our method across the axes of sampling speed, error rate and FID demonstrates a conceptual step beyond the previous state-of-the-art for compositional generation. We further show that our approach can be applied to an out-of-the-box pre-trained text-to-image model to allow for principled and controllable generation without any fine-tuning. Though outside the scope of our present work with controllable image generation, the prospect of applying our method for other compositional tasks (such as multi-prompt text generation) remains an intriguing possibility for future work. Altogether, we believe our work provides a strong foundation for future work in the direction of controllable image generation with composed parallel token prediction.

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

## APPENDIX

## A FULL DATASET DETAILS

Below we provide a full description of each dataset used in the quantitative evaluation of our method.

- **FFHQ** (30): FFHQ is a dataset of 70,000 aligned images of human faces. Three binary attribute labels are available for each image: "smile"/"no smile", "glasses"/"no glasses", "male"/"female", which we use to condition the generation process.

- **Positional CLEVR** (29): CLEVR is a synthetic dataset of rendered 3D objects of various colours, shapes, sizes and textures. In the Positional variant of CLEVR, object attribute and position annotations are available. Following (7), we use a 30,000-image subset of CLEVR (restricted to contain between 1 and 5 objects per image). For this task, image generation is conditioned on object position only.

- **Relational CLEVR** (29; 7): Relational CLEVR is similar in appearance to Positional CLEVR, with the addition of text annotations for objects and their relationships (e.g. "*the red cube is above the blue sphere*"). Image generation is conditioned on (tokenized) text descriptions of object attributes and relationships, including object shape, size, material, colour, and relative position.

## B QUANTITATIVE RESULTS FOR RELATIONAL CLEVR AND FFHQ

Tables 2 and 3 contain the results of our quantitative experiments (error rate and FID) for the Relational CLEVR and FFHQ Datasets. These are discussed in the main text but omitted for brevity.

## C ERROR VS FID PLOTS

Figures 7, 8 and 9 are scatter plots of error rate against FID corresponding to results in Tables 1, 2 and 3. Included on the same axes of each plot are the empirical Pareto front of the data, which we

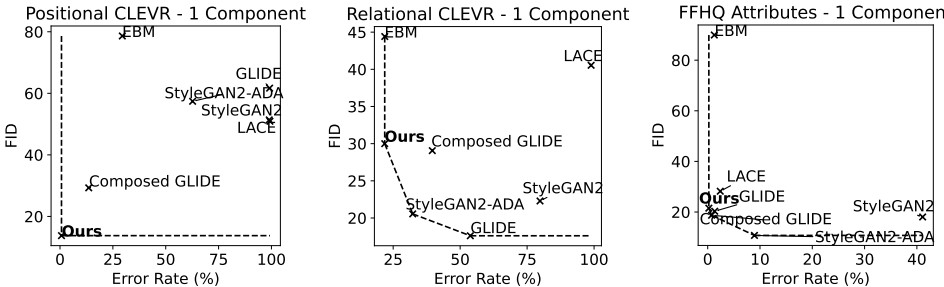

Figure 7: Scatter plots of compositional generation error vs FID on 3 datasets (1 input component): Our method lies on the Pareto front of all results while achieving state-of-the-art (lowest or joint lowest) error.

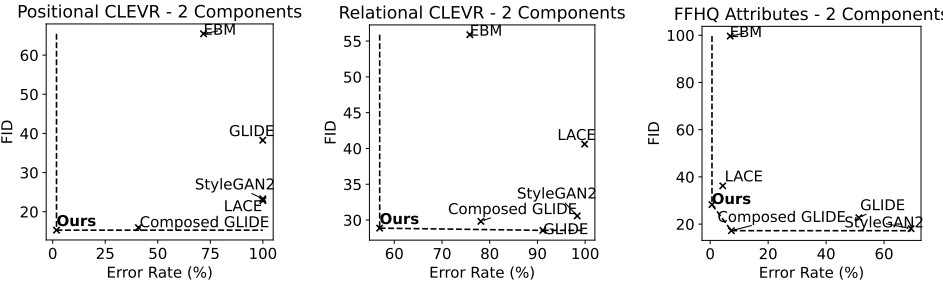

Figure 8: Scatter plots of compositional generation error vs FID on 3 datasets (2 input component): Our method lies on the Pareto front of all results while achieving state-of-the-art (lowest or joint lowest) error.

define as the unique linear piece-wise form that is convex, monotonic, and is as close as possible to the lowermost and leftmost (best) of the data points while extending to the top and right of each plot. Our results all lie on, or very close to, the empirical Pareto front of their respective plots while having the lowest error rate when ranked among the baselines.

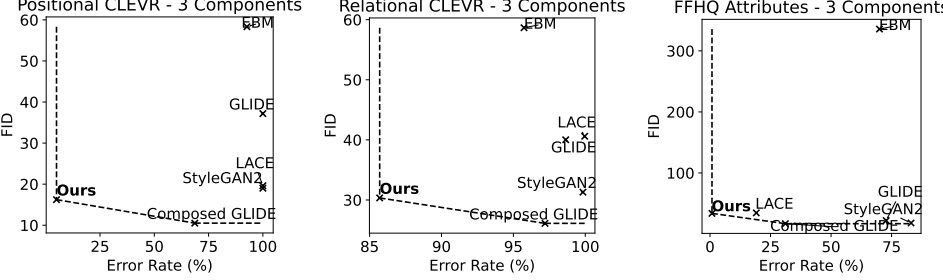

Figure 9: Scatter plots of compositional generation error vs FID on 3 datasets (3 input component): Our method lies on the Pareto front of all results while achieving state-of-the-art (lowest or joint lowest) error.

## D DERIVATIONS

Here we include additional derivations which were omitted from the main paper for brevity.

## D.1 CANCELLING THE $P(\boldsymbol{c}_i)$ TERMS

The following is a full derivation of the result given in (2) of the main text.

$$P(\boldsymbol{x} = x_j | \boldsymbol{c}_1, ..., \boldsymbol{c}_n) = \frac{P(\boldsymbol{x} = x_j) \prod_{i=1}^{n} \frac{P(\boldsymbol{x}=x_j|\boldsymbol{c}_i)P(\boldsymbol{c}_i)}{P(\boldsymbol{x}=x_j)}}{\sum_{u=1}^{k} \left[ P(\boldsymbol{x} = x_u) \prod_{i=1}^{n} \frac{P(\boldsymbol{x}=x_j|\boldsymbol{c}_i)P(\boldsymbol{c}_i)}{P(\boldsymbol{x}=x_j)} \right]} \tag{8}$$

$$= \frac{P(\boldsymbol{x} = x_j) \prod_{i=1}^{n} \frac{P(\boldsymbol{x}=x_j|\boldsymbol{c}_i)}{P(\boldsymbol{x}=x_j)}}{\sum_{u=1}^{k} \left[ P(\boldsymbol{x} = x_u) \prod_{i=1}^{n} \frac{P(\boldsymbol{x}=x_u|\boldsymbol{c}_i)}{P(\boldsymbol{x}=x_u)} \right]} \tag{9}$$

In this way, the contribution of $P(\boldsymbol{c}_i)$ terms is cancelled out, since $P(\boldsymbol{c}_i)$ are constant with respect to different values of $x_j$. In subsequent sections, we show how this brief but important result for conditional categorical distributions can be successfully extended to parallel token prediction for image generation, achieving state-of-the art error rate and speed on 3 datasets.

## D.2 EXTENSION OF PRODUCT OF EXPERTS TO SEQUENTIAL DISCRETE GENERATION

Here we clarify the relationship between equations (2) and (3) in the main text.

Equation (3) holds if and only if the joint distributions $s_t, c_i$ and $s_t, c_i$ are independent conditional on $s_{t+1}$, i.e. $P(s_t, c_i, c_j | s_{t+1}) \propto P(s_t, c_i | s_{t+1}) P(s_t, c_j | s_{t+1})$ for all pairs of distinct input conditions $c_i, c_j$ (this is the product of experts assumption, extended to sequential conditional generation).

$P(s_t, V | s_{t+1}) = P(V | s_{t+1})$ for any random variable or collection of variables $V$, since $s_t$ is entirely determined by $s_{t+1}$ since the generation process is purely additive, meaning the representation of $s_t$ is contained by that of $s_{t+1}$. Therefore the expression of the extended product of experts assumption can be simplified to:

$$P(c_i, c_j | s_{t+1}) \propto P(c_i | s_{t+1}) P(c_j | s_{t+1})$$

i.e. the extended product of experts assumption is equivalent to the original product of experts assumption and therefore (3) follows from (2) with the substitution of a categorical variable $x$ with a stateful discrete generative process $s_{t+1} | s_t$.

## D.3 COMPOSITIONAL NEXT-TOKEN PREDICTION

In the main text we claim that our discrete compositional method can be applied to arbitrary generative methods provided they are discrete and iterative. Here we back up this claim by showing how our method can be adapted to autoregressive (next-token) sampling.

In the specific case of discrete autoregressive image modelling, a single latent code (token) is generated at each time step, conditioned on some initial context in addition to previously generated tokens. In this situation, each successive state $\boldsymbol{s}_{t+1}$ is simply the concatenation of the previous state $\boldsymbol{s}_t$ and the subsequent generated token $\boldsymbol{x}_{t+1}$. Thus the random variable $\boldsymbol{s}_{t+1}$ can be restated as:

$$\boldsymbol{s}_{t+1} = \boldsymbol{s}_t \oplus \boldsymbol{x}_{t+1} \tag{10}$$

where $\oplus$ denotes the concatenation of two tokens or strings of tokens. Consequently, the conditional generation task in (3) can be reformulated in terms of sampling the next token given the previous state:

$$P(\boldsymbol{x}_{t+1} | \boldsymbol{s}_t, \boldsymbol{c}_1, ..., \boldsymbol{c}_n) \propto P(\boldsymbol{x}_{t+1} | \boldsymbol{s}_t) \prod_{i=1}^{n} \frac{P(\boldsymbol{x}_{t+1} | \boldsymbol{s}_t, \boldsymbol{c}_i)}{P(\boldsymbol{x}_{t+1} | \boldsymbol{s}_t)} \tag{11}$$

i.e. only a single token is considered at each generation step, making our formulation compatible with autoregressive (next-token) prediction. In practice, we speculate that the autoregressive case is less compatible with our compositional method than parallel token prediction due to being less strongly

regularised (and hence more prone to over-fitting: parallel token prediction is strongly regularised by design (10)), in addition to being more sensitive to the accumulation of errors due to tokens being generated "left-to-right, top to bottom" in the image. Furthermore, there is no guarantee that autoregressive models provide a calibrated estimate of conditional/unconditional probabilities, which may further limit hypothetical performance. For these reasons, we maintain our focus on parallel token prediction which is found to to outperform the previous state-of-the-art on image generation error rate when applied alongside our discrete composition method.

### D.4 OMISSION OF THE DISJUNCTION (OR) OPERATOR

While the implementation conjunction ("AND") and negation ("NOT") operators are highly effective for controllable generation, they do not correspond exactly to Boolean algebra: in our framework, negation is distributive, i.e. $-(a + b) \equiv -a - b$, but in Boolean algebra, it is not: NOT($a$ AND $b$) $\neq$ NOT $a$ AND NOT $b$ for $a \neq b$. For this reason, our framework does not allow for the straightforward implementation of the disjunction (OR) operator, which we do not explore in our qualitative or quantitative results.

## E ADDITIONAL QUALITATIVE EXPERIMENTS

### E.1 CONCEPTUAL PRODUCT SPACE

Fig. 10 illustrates how our compositional method can be used to generate a "product space" over visual concepts. In particular, Fig. 10 demonstrates the Cartesian product of the "colour" concept {"a red object","an orange object",...} with the "category" concept {"a cat","a dog","an apple","a cherry"} using our approach. Concept weights $w_1$ and $w_2$ are set at 6 for all samples. We used the aMUSEd (13) implementation of MUSE (11) (text-to-image masked generative transformer) as the pre-trained backbone model, as with other qualitative text-to-image experiments.

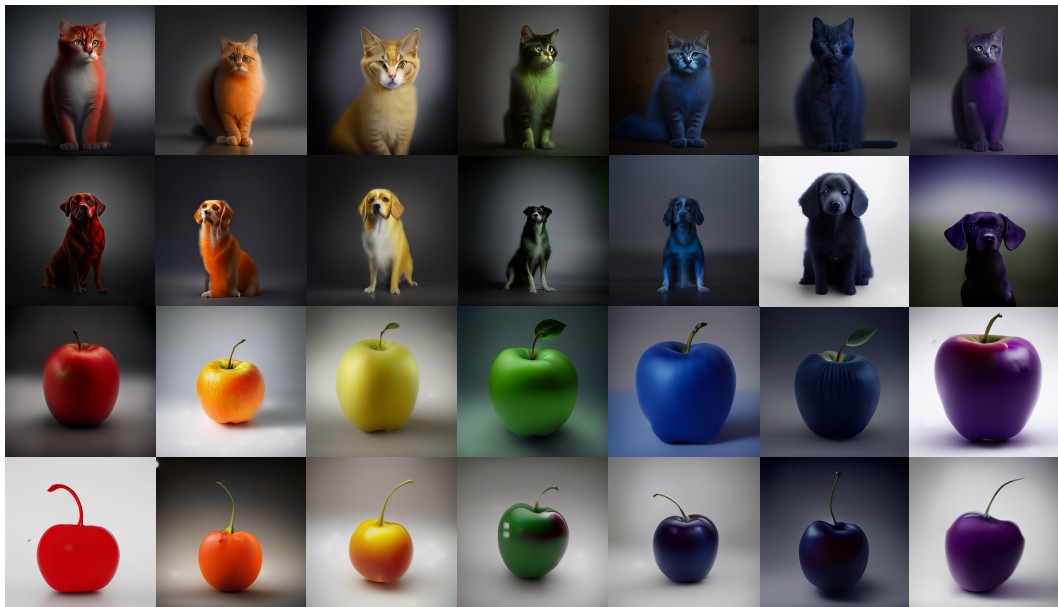

Figure 10: Conceptual product space: Example of composing two concept spaces using our framework: {"a cat","a dog","an apple","a cherry"} $\times_{\textbf{AND}}$ {"a red object","an orange object","a yellow object","a green object","a blue object","an indigo object","a violet object"}.

### E.2 VARYING CONCEPT WEIGHT FOR FFHQ

In the main text we visualise the effect of varying $w_{male}$ for the FFHQ dataset. Figures 11 and 12 visualise the effect of varying the weights of the remaining two concepts in FFHQ ($w_{male}$ and $w_{smile}$ respectively).

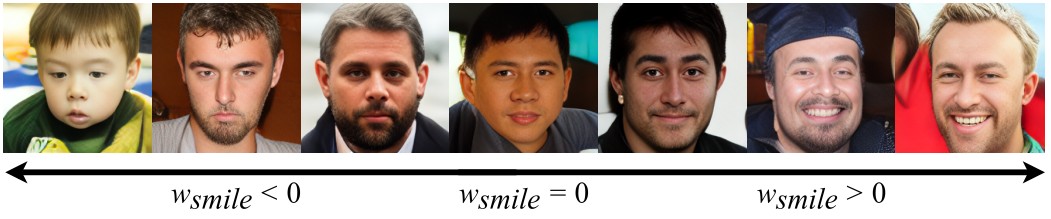

Figure 11: Effect of varying concept weight $w_{male}$ for our model trained on the FFHQ dataset: Concept weighting (both positive and negative) allows for fine-grained control of output attributes.

### E.3 TEXT-TO-IMAGE COMPARISON WITH BASELINE

In order to give more context to the results in Figure 3 in the main text, in Figure 13 we compare against the single-prompt baseline. The single prompts were constructed by concatenating the input prompts with "; " as a delimiter. We find in certain cases that the baseline model omits omits details (e.g. the owl in the first example) while applying adjectives to the wrong nounrs (e.g. "pink nose" is applied to the entire cat in the fourth example; "reflected" is applied to the boat and not the mountain in the final example). Our method does not suffer from these issues, indicating greater controllability.

Figure 12: Effect of varying the $w_{no\_glasses}$ concept weight from $-3.0$ to $3.0$ while keeping $w_{male} = w_{no\_glasses} = 3.0$.

## F FULL MODEL TRAINING DETAILS

Here we give the full technical details of our training runs in addition to hardware considerations.

For each dataset, we utilise a deep residual convolutional vector-quantized autoencoder architecture following the protocol of (9) and (10) which have been previously shown to produce high-fidelity reconstructions for a variety of image datasets. We use a training batch size of 20 for CLEVR and Relational CLEVR, with a smaller batch size of 8 for FFHQ due to the larger image size. CLEVR and Compositional CLEVR VQ-VAE models are trained for $20,000$ iterations each, while the FFHQ VQ-GAN is trained for $100,000$ iterations due to the smaller batch size and larger resolution, with adverserial loss starting at $30,000$ iterations.

We set the VQ-VAE/VQ-GAN codebook size at 256 for all three datasets, which we find to be sufficient for obtaining high-quality reconstructions. We use the encoder-only transformer architecture, using the exact same same architecture for all three datasets (from (10)) (24 layer, embedding dimension 512, fully connected hidden dimension 2048). We train a total of 3 samplers (one for each dataset) for $300,000$ iterations each. Training each sampler model took 1.5 days per model, with an additional 0.5 days for evaluations. Training of all models and running evaluation took approximately 5 days in total, using a single NVIDIA GeForce RTX 3090 and Pytorch implementations (see code in supplementary). Preliminary and failed experiments (e.g. due to bugs) made up for around 3 days of compute on the same hardware.

For each dataset, we encode input conditions as additional embeddings which are concatenated to the latent embeddings before being fed to the transformer. Object position for CLEVR is encoded using a learned linear map $M_{pos} : \mathbb{R}^2 \rightarrow \mathbb{R}^d$ where $d$ is the hidden dimension of the transformer. Face attributes for FFHQ are encoded using learned embedding $M_{face}$ :

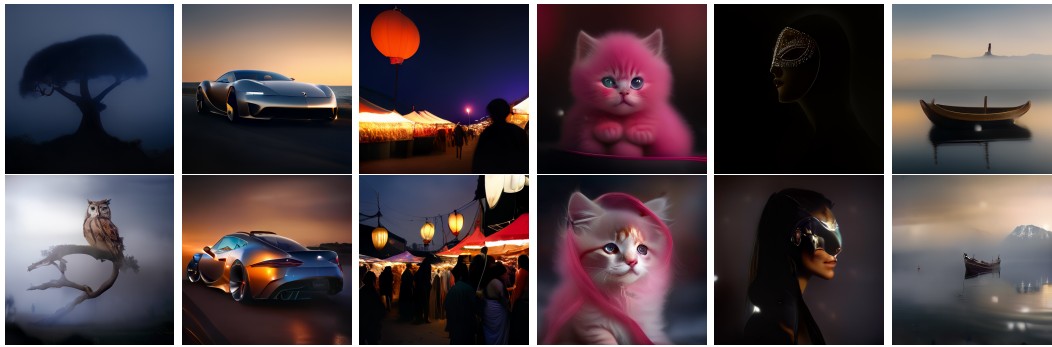

"Ancient oak tree with wide trunk and expansive canopy" **AND** "Owl perched on branch" **AND** "Mist swirling around base of tree"

"Sleek, futuristic sports car with metallic finish" **AND** "Car racing along winding coastal highway at sunset" **AND** "Lens flare from setting sun on car's surface"

"Vibrant, bustling outdoor market with colorful stalls" **AND** "Vendors and shoppers from various cultures interacting" **AND** "Hanging lanterns illuminating the scene"

"Fluffy kitten with big eyes and pink nose" **AND** "Kitten tangled in ball of red yarn" **AND** "Soft lighting casting gentle shadows"

"Woman's face in profile, delicate features" **AND** "Bejeweled masquerade mask covering upper half of face" **AND** "Contrasting lighting, one side illuminated, other in shadow"

"Rustic, wooden rowboat floating on misty lake" **AND** "Snow-capped mountains in the distance, reflected in water" **AND** "Morning light filtering through mist, golden glow"

Figure 13: Compositional text-to-image results with captions (zooming recommended). *Top:* single-prompt CFG baseline. *Bottom:* composed multi-prompt (ours). Our framework allows for the composition of multiple conditions, conferring an advantage over the single-prompt baseline.

{"smile", "no smile", "glasses", "no glasses", "female", "male"} $\rightarrow \mathbb{R}^d$. For Relational CLEVR, we tokenize text descriptions, mapping to learned token embeddings and adding positional embeddings before concatenating with the learned image token embeddings (also adding learned position embeddings for the image token embeddings).

## G BASELINES DETAILS

In the quantitative evaluations of our method, we compare against the baseline results provided in (7) in addition to the results of the method in (7). For completeness, below we provide a brief description of how the baseline results were obtained in (7):

- **StyleGAN2-ADA** (33) - The StyleGAN2-ADA results were obtained in (7) using the off-the-shelf model provided by (33).

- **StyleGAN2** (34) - Compositional StyleGAN2 results were obtained in (1) by training classifiers on the latent space of StyleGAN2, which were then used to generate novel latent representations. StyleGAN2 models were either used off-the-shelf (for FFHQ) or trained from scratch (for Positional and Relational CLEVR).

- **LACE** (5) - LACE results were obtained in (7) by composing energy-based models acting on the latent space as in (5), with training data generated by StyleGAN2-ADA (above).

- **GLIDE** (35) - The (non-composed) GLIDE results were obtained in (7) by encoding input conditions as a single, long sentence, with outputs being upsampled separately from $64 \times 64$.

- **EBM** (6) - Composed EBM results were obtained in (7) by composing conditional energy functions for multiple concepts as in (6).

- **Composed GLIDE** (7) - Composed GLIDE results were obtained by (7) using the method for composing diffusion outputs proposed by (7).

We had insufficient resources to train our own models from scratch, and so we instead precisely replicated the evaluation protocol of (7) to enable fair comparison with our own method.

## H    BINARY CLASSIFIER DETAILS

In the main text we include generation error rate metrics for baseline methods (from (7)) and our own experiments with Positional CLEVR, Relational CLEVR and FFHQ. Here we document fully how we obtained our own error rate scores by following the evaluation approach of (7) so as to maintain valid comparisons with the baseline results reported by (7).

Accuracy is determined by a binary classifier for each of the three datasets, which takes both an image and an attribute as input and produces a binary output corresponding to whether the specified attribute is present. We obtained the error rate scores following the exact same evaluation approach of (7) so as to maintain valid comparisons with the baseline results reported by (7). For each experiment we generate $5000$ images, computing accuracy (Acc) and FID for each group of $5000$. Samples are taken over 30 time steps (corresponding to unmasking $256/30 \approx 8.53$ tokens per time step on average). We fix the concept weight $w_i$ at 3.0 for all experiments. We detail all quantitative results in Table 1 and in Tables 2 and 3 in Appendix B (best performance is written in bold for each column, second-best is underlined).

For CLEVR and Relational CLEVR, we use the pre-trained classifiers provided by (7), which have validation classification accuracy scores of $99.05\%$ and $99.80\%$ respectively. For FFHQ, since no pre-trained classifier was available we trained binary classifiers following the same procedure as (7) (one for each attribute, with a $80:20$ train-validation split). The classifiers achieve equal or greater validation accuracy than the classifiers usef by (7). The high validation accuracy scores for the evaluation classifiers are deemed sufficient to allow reliable estimation of the error rate for our generated images. Our quantitative evaluations follow the exact same procedure used to obtain the baseline results (7), allowing for a fair comparison with the baselines.

## I    STANDARD UNCERTAINTY COMPUTATION

In Tables 1, 2 and 3 we include standard uncertainties at two standard deviations ($2\sigma$). We base this computation on the number of sampled images in order to give context to the difference between baseline results and our own results (this is especially useful when results are close together). We are unable to report error bars based on multiple repeats of training runs because such error bars were not reported in (7) and we lack the resources to perform our own runs of their experiments. For these reasons, the value of $2\sigma$ (two standard deviations) is derived and computed as follows for a percentage accuracy score $p$:

We assume that a given method generates an image correctly (consistent with the specified conditions) with probability $p$, independently for each of $n$ trials (generated samples). It follows that the number of "correct" samples $X$ is distributed according to the Binomial distribution:

$$X \sim B(n, p) \tag{12}$$

The variance in the number of correct samples $X$ is then:

$$Var(X) = np(1 - p) \tag{13}$$

And thus the standard deviation is:

$$SD(X) = \sqrt{np(1 - p)} \tag{14}$$

The standard deviation $\sigma$ of the accuracy score (or equivalently, the error rate) is then $SD(X)$ divided by $n$:

$$\sigma = \frac{\sqrt{np(1-p)}}{n} \tag{15}$$

$$= \frac{\sqrt{p(1-p)}}{\sqrt{n}} \tag{16}$$

$$= \sqrt{\frac{p(1-p)}{n}}(\times 100\%) \tag{17}$$

$$\tag{18}$$

Finally, we clip values for $2\sigma$ to be no greater than $p$ and no greater than $1 - p$ so as to avoid giving error bounds greater than $100\%$ or smaller than $0\%$. We compute $2\sigma$ in the same way for all accuracy results (including those reported by (7)) since they are all computed based on $5000$ generated samples.

We omit uncertainties for FID for two reasons: (1) the uncertainty in FID for comparing two sets of 5000 images is expected to be low (32) and (2) it would take too long with our available computational resources to compute these by repeating all experiments (including running the baselines) multiple times.

## J  Nearest Neighbours of Generated Samples

To verify that our method does not simply reproduce samples from the training data (over-fitting), for each dataset we generate a batch of 8 images based on a random selection of 3 input conditions (positions, relations and attributes for CLEVR, Relational CLEVR and FFHQ respectively). In this experiment, we choose to study the composition of 3 input conditions (as opposed to 1 or 2) as this situation is the most likely to produce over-fit images (due to it finding the "smallest", or lowest-entropy section of the sample space). We compute the 8 nearest neighbours of each sample from the original training data based on perceptual distance. Figures 14, 15 and 16 visualise the results. The leftmost column of each figure contains the (non-cherry-picked) generated samples, while the remaining 8 images in each row are the nearest neighbours. These figures show qualitatively that none of the 8 generated samples from each dataset perfectly match the nearest neighbours, indicating strong generalisation performance. All samples were taken at temperature 0.9 and condition weight 3.0, in accordance with quantitative experiments in the main text.

## K  Code License and Running Instructions

Please see LICENSE in the code for full license(s).

Please see README.md in the code for running instructions for reproducing our quantitative experiments, including model training, classifier training, and obtaining results.

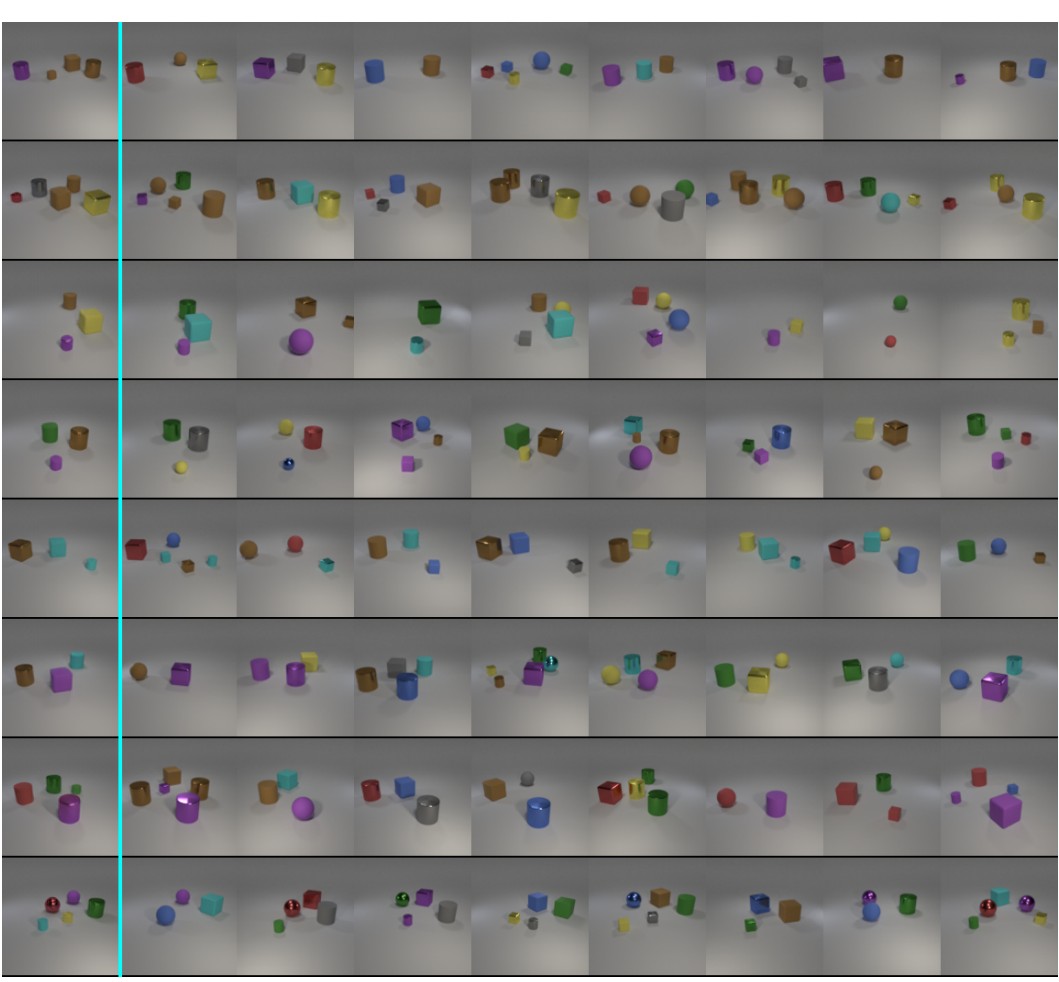

Figure 14: Perceptual nearest neighbours for the (positional) CLEVR dataset. Leftmost column is (non-cherry-picked) generated samples, remaining images in each row are the 8 nearest neighbours (left to right goes from nearest to farthest).

.

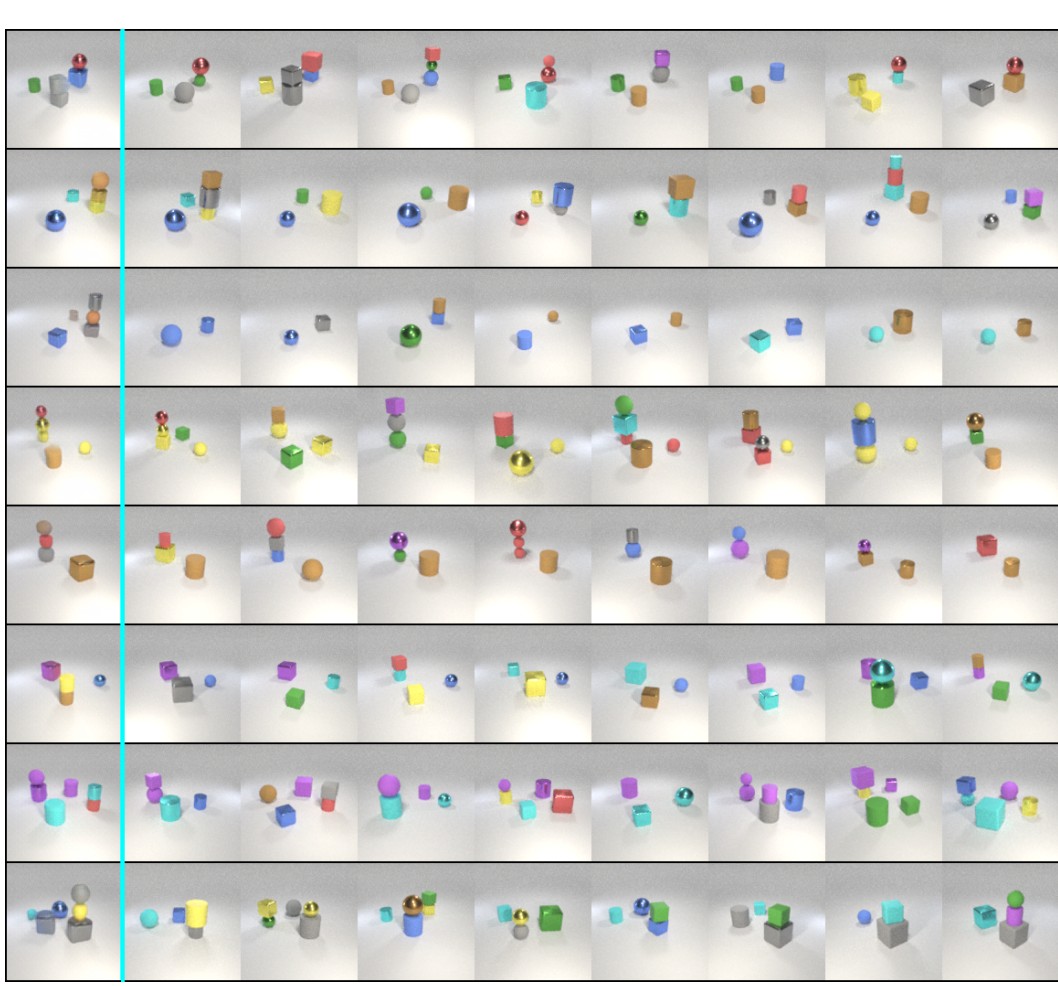

Figure 15: Perceptual nearest neighbours for the Relational CLEVR dataset. Leftmost column is (non-cherry-picked) generated samples, remaining images in each row are the 8 nearest neighbours (left to right goes from nearest to farthest).

.

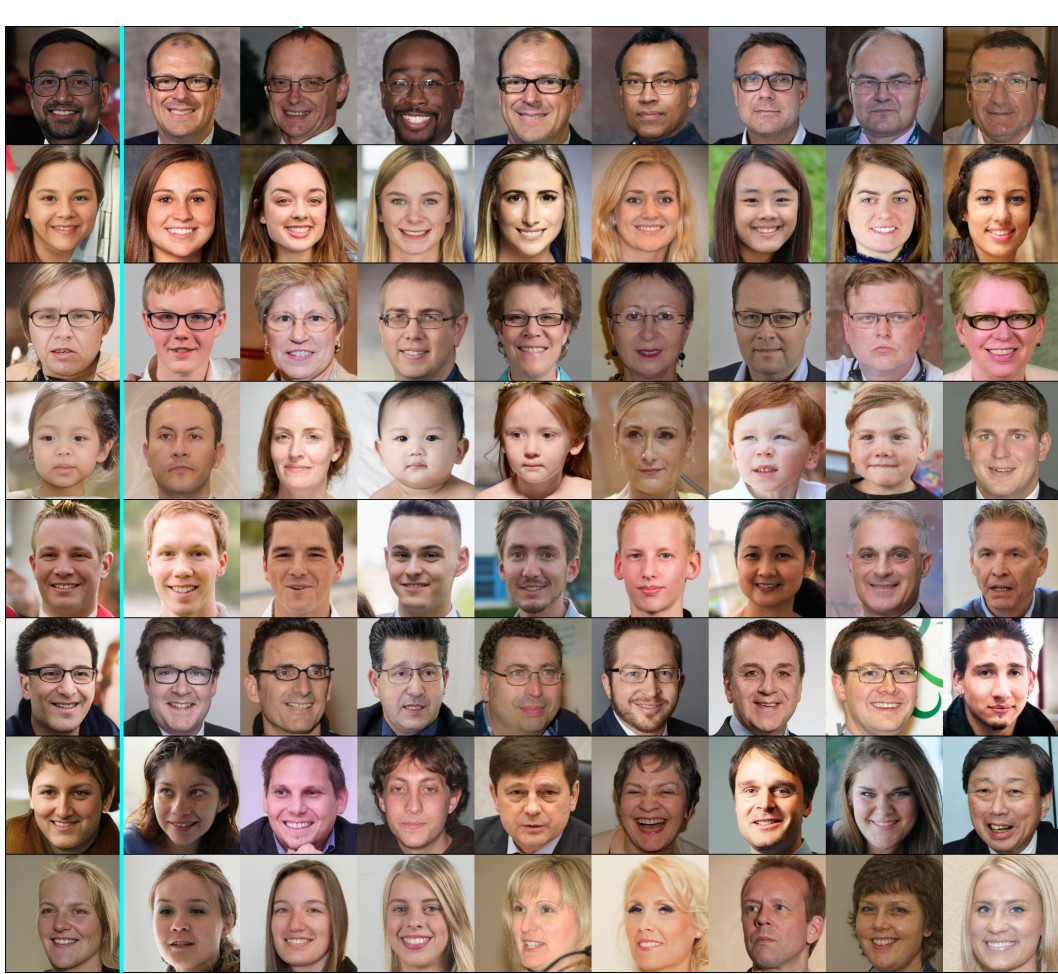

Figure 16: Perceptual nearest neighbours for the FFHQ dataset. Leftmost column is (non-cherry-picked) generated samples, remaining images in each row are the 8 nearest neighbours (left to right goes from nearest to farthest).

.

