# OpenReview forum: "Compositional VQ Sampling for Efficient and Accurate Conditional Image Generation"
_ICLR.cc/2025/Conference — Submitted to ICLR 2025_

### Official Review · Reviewer_fyVr · 2024-11-04

**Soundness:** 3
**Presentation:** 3
**Contribution:** 2
**Rating:** 5
**Confidence:** 3

**Summary:**

The paper presents a compositional approach for image generation. The authors proposes to incorporate multiple conditions within a discrete masked transformer generation. Under the assumption that these conditions are independent, each condition contributes to the discrete denoising of the masked latent space at each diffusion time.
 The conditions are shown to control the specific attributes, both positively and negatively, as well as generate high quality images both quantitatively and qualitatively. In addition, the method is shown to be applicable to pre-train t2i discrete diffusion namely aMUSEd).

**Strengths:**

- The paper is well written and easily understood
- The results support the claims
- The conditions are able to generalize to OOD sampling

**Weaknesses:**

My main concern is the novelty of this paper.
Masked diffusion transformers are not new as noted by the author it is used for example in aMUSEd.
In addition compositional conditions was presented in "Compositional Visual Generation with Composable Diffusion Models" which derive the same conjunction and negation attributes.  While their are some differences (the use VQ-VAE vs VQGAN) the method can be seem as a compositional aMUSEd, which is quite limited in novelty.
I wish the authors to better explain what exactly this work proposes that aMUSEd and compositional conditions did not.

- The conditions are entangled with other attributes as can be seen in Fig 6.
- Each condition requires a forward pass, in an already slow diffusion generation process.
- Low resolution image generation.

**Questions:**

- Please address the novelty concern and other weakness mentioned in the weakness section.

---

> ### Author Response · Authors · 2024-11-20
> **Response to reviewer**
>
> We thank the reviewer for their careful assessment of our work. Below we aim to clarify points where appropriate.
>
> In direct response to each of the bullet points raised in "weaknesses", in the order of appearance:
> 1. A stronger condition weighting increases confidence while reducing diversity, akin to a temperature parameter, and therefore is more sensitive to mode collapse (as in the sunglasses case). This is true of any temperature parameter in probabilistic sampling. The entanglement of (sunglasses/reading glasses) with the condition weighting merely indicates a higher prevalence of sunglasses in the dataset compared to reading glasses. In the limit $w\rightarrow\infty$, we would expect the method to always output the same image, as would be the case of setting temperature to 0 in any analogous situation.
> 2. The speedup relative to the baseline compositional methods ($2.3-12\times$) compensates for the requirement of additional forward passes ($~3\times$), which is on a comparable order of magnitude. The requirement of additional forward passes is stated in the paper as a known and necessary limitation, and in our view this is not a weakness of the contribution itself.
> 3. We agree that it would be nice to show our approach outperforming the baselines at greater resolution in addition to the resolutions shown. Besides increased compute requirements, there is no fundamental property of our method which limits its application to 128x128, 256x256 (c2i) or 512x512 images (t2i). However, we had limited compute for our experiments which we exploited to the greatest degree possible, and in doing so, we believe we have adequately demonstrated the benefits and potential impact of our method.
>
> Regarding the reviewer's observation that "the method can be seen as a compositional aMUSEd": we suggest that terming our method "compositional aMUSEd" is potentially unfaithful to the origins of generative masked transformers, which pre-date aMUSEd by around 3 years. Only part of our contribution (that of adapting a pre-trained masked generative transformer model for compositional text-to-image generation without any fine-tuning) relates to aMUSEd.
>
> Other reviewers have commented on our paper's novelty as a strength. We have formally derived why our method for compositional image generation works in theory. Our derivation is novel for discrete processes, and accordingly requires theoretical considerations above, beyond and independent of those of continuous processes, to which we dedicate over 2 full pages of our paper (spanning pages 4-6). We show empirically that the method does in fact work consistently with the theory, and not only this, but it is dramatically faster (2.3-12 times) and dramatically more accurate (cutting error by 63% on average) than a strong continuous baseline, all while having comparable parameter efficiency and FID scores. We hope this clarifies the novel aspects of our work and its potential impact, and if not, we are very happy to continue discussion until the end of the discussion period.

---

> ### Author Response · Authors · 2024-11-26
>
> Our earlier comment aimed to respond fully and faithfully to each of the reviewer's concerns. We would like to check with the reviewer whether we have adequately addressed these concerns, before this stage of the discussion period ends?

---

> ### Author Response · Authors · 2024-11-28
>
> We would like to gently follow up from our previous message to check if the reviewer feels we have adequately addressed the concerns/questions, and if so if the reviewer might kindly increase their rating as a result?

---

### Official Review · Reviewer_7koU · 2024-11-04

**Soundness:** 3
**Presentation:** 3
**Contribution:** 2
**Rating:** 5
**Confidence:** 4

**Summary:**

The paper investigates multiple conditioning approaches for the Masked Generative Image Transformer, focusing on the composability of multiple predictions. The authors demonstrate that using multiple conditioning signals improves control over the generation process. Their approach shows substantial improvements across three datasets, significantly reducing the error rate in conditional image generation. Additionally, they present interesting properties for negative conditioning and out-of-distribution generation for text-to-image tasks.

**Strengths:**

- The paper demonstrates strong results in terms of error rate reduction, highlighting effective conditioning.
- The method leverages the MaskGIT framework, achieving high throughput (compared to diffusion models) while maintaining a low parameter count.
- The paper also reveals interesting properties for out-of-distribution (OoD) synthesis and negative prompting

**Weaknesses:**

- The contribution of conditioning the model alone is already well-established through Classifier-Free Guidance (CFG). Applying this technique for multiple conditions for discrete spaces may be a weak contribution.
- While the manipulation of text-to-image conditioning appears promising, the lack of quantitative results makes it difficult to fully assess its effectiveness. I think that the complete Figure 13 in the main paper would enhance clarity.
- Similar to other methods (e.g., [7]), this approach requires additional forward passes, which slow down the overall pipeline.
- The method shows a relatively weak FID score on the FFHQ dataset.

**Questions:**

- Why does the FID score change when the number of components varies that much (particularly for Table 3)?
- For out-of-distribution synthesis, under what conditions does the model fail? Can it generate more than eight objects?
- What happens in the case of synthesizing both 𝑐  and −𝑐 ?
- Is it necessary to maintain all conditioning signals until the end of the generation process? For instance, on FFHQ, conditioning on glasses for the first few iterations and switch to no condition?
- Is the current format for references correct?

---

> ### Author Response · Authors · 2024-11-20
> **Response to reviewer and details of revision**
>
> We thank the reviewer for their constructive questions and suggestions for improvements. We aim to address each of these points below, and we respond to suggested improvements in the most recent revision of our paper (see points below for specific changes).
>
> Addressing the points under "weaknesses" in-order:
> 1. We have dedicated more than two pages to the derivation of compositional conditional generation for masked generative models. Without this derivation (one of our core contributions), there would not have been sufficient motivation to implement, train and extensively evaluate our approach (another of our core contributions). We include this derivation in detail so as to elevate the quality of the presented work by providing a rigorous motivation to our approach. The reviewer states that this "may be a weak contribution", so we would like to draw attention to:
>    * The state-of-the art results for accuracy and efficiency while maintaining competitive FID, across 3 very different experimental settings, compared to already strong continuous compositional baselines
>    * The capacity of our method to be applied to an existing pre-trained T2I model without any additional fine-tuning for improved control over T2I outputs
>    * The lack of any similar derivation, implementation, evaluation or presentation of results in the literature as of the time of submission
>
>    We hope the reviewer will kindly revise their assessment of our contribution in light of these clarifications.
>
> 2. We thank the reviewer for this suggestion: We agree that the full Figure 13 will enhance clarity if placed in the main text. This is now present in the most recent revision (top of page 3) which was possible without going over the 10-page limit.
> 3. The speedup relative to the baseline compositional methods ($2.3-12\times$) compensates for the requirement of additional forward passes ($~3\times$), which is on a comparable order of magnitude. The requirement of additional forward passes is stated in the paper as a known and necessary limitation and is not a weakness of the work itself in our view.
> 4. The reviewer's statement that FID is "relatively weak" may be imprecise or misleading in the context of the improved accuracy. We are keen to point out that the FID score on FFHQ is poor _relative only to baselines which achieve considerably higher error rate_, indicating reduced diversity but considerably greater control over the output image. Compared to the next-best baselines in terms of error rate on FFHQ, our model achieves comparable (1 component) or better (2 and 3 components) FID (Table 3, main text). This is an average 83.9% reduction in error rate on FFHQ while still reducing FID by 3.2% on average.
>
> In response to the questions:
> 1. The introduction of more input components restricts the sample space, reducing diversity in favour of control. This explains the increase in FID and is a direct consequence of the probabilistic formalism of our method. Despite this increase in FID in Table 3, our method still attains lower FID scores than the method with the next-lowest error rate for 2 out of 3 input component settings for FFHQ.
> 2. We found that up to and including 8 objects are reliable for OOD synthesis of Positional CLEVR, with 9+ objects failing more with the addition of more input conditions. This is expected, since errors are expected to accumulate with the addition of conditions (residual variance is additive).
> 3. Since -c and c are not independent attributes (they are mutually exclusive), the independence assumption is explicitly violated. The result is that the contributions perfectly cancel out, thus generating an image as if the condition c or -c were not specified (i.e. P(x) as opposed to P(x|c) or P(x|-c)).
> 4. This is a very interesting and insightful question which, while beyond the scope of our contribution, we would be excited to explore in future work. Carefully observing the effect of dropping the input conditions after a few time steps could further improve generation efficiency with minimal effect on performance, e.g. since the number of sunglasses-specific tokens may already be sufficient in the partial image to serve as cues for a full and accurate image.
> 5. We used the official ICLR2025 LaTeX template for everything. We believe the presentation of references to be correct, but we welcome any specific correction if this is not the case, which we will action as soon as possible in a subsequent revision.

---

> ### Author Response · Authors · 2024-11-26
>
> Our earlier comment and revised submission aimed to respond in detail to each of the reviewer's concerns, recommendations and question (see also: revised paper). We would like to check with the reviewer whether we have adequately addressed these concerns, before this stage of the discussion period ends?

---

> > ### Comment · Reviewer_7koU · 2024-11-26
> >
> > I would like to thank the authors for their thoughtful and detailed responses, and I appreciate the time and effort they devoted to addressing the feedback.
> >
> > - While the work is interesting, I find the novelty somewhat limited, , as noted by all the reviewers (EQrG, fyVr, qGWC) and myself.
> > - Additionally, the results on the text-to-image task could be further strengthened to maximize the paper's impact.
> > - Although the method demonstrates a significant improvement in error rate, the FID on FFHQ (the only "high-resolution" dataset included) remains too high.
> >
> > For these reasons, I decided to keep my score of 5.
> >
> > As a side note, I observed that the references in your paper are formatted numerically, in contrast to the "Name et al., Date" style used in other papers I reviewed.

---

> > > ### Author Response · Authors · 2024-11-28
> > >
> > > Firstly we would like to thank the reviewer for their in-depth assessment of our work and subsequent engagement.
> > >
> > > In response to each of the most recent points:
> > >
> > > 1. In our view, the state-of-the-art results speak to the novelty of our method. Were it not novel, it would be unable to cut error rate so significantly while also increasing efficiency. We would also like to emphasise that we haven't merely tweaked an existing approach to cut error rate: this is the first application of masked generative transformers to compositional generation, to our knowledge.
> > > 2. The text-to-image task is a demonstration of the flexibility of our approach, it is not part of our quantitative evaluation, and in our view, it does not need to be in order to demonstrate the potential impact of our method. We demonstrate the utility of composition and negation in gaining greater control over text-to-image outputs.
> > > 3. Quoting our previous response:
> > >    > We are keen to point out that the FID score on FFHQ is poor _relative only to baselines which achieve considerably higher error rate_, indicating reduced diversity but considerably greater control over the output image. Compared to the next-best baselines in terms of error rate on FFHQ, our model achieves comparable (1 component) or better (2 and 3 components) FID (Table 3, main text). This is an average 83.9% reduction in error rate on FFHQ while still reducing FID by 3.2% on average.
> > >
> > >    To paraphrase: There is no method in the literature which gets _both_ low FID _and_ low error rate on FFHQ (see baseline results, Table 3). Our method demonstrates a very reasonable trade-off, cutting error rate over the next-best baselines **significantly** with **very little change** in FID relative to those same baselines.
> > >
> > > For these reasons, we kindly ask the reviewer to increase their rating. We have made changes and clarifications as the reviewer requested under "weaknesses" and we have directly answered the reviewer's questions.

---

### Official Review · Reviewer_EQrG · 2024-11-04

**Soundness:** 3
**Presentation:** 3
**Contribution:** 2
**Rating:** 6
**Confidence:** 2

**Summary:**

The paper introduces a new approach for compositional image generation for discrete generative processes, leveraging a compositional code sampling framework applied to discrete spaces with Vector-Quantization technique, and masked token prediction method for efficient image generation. The proposed method combines log-probabilities of discrete generative models, allowing control over compositional generation conditions without specialized training loss. Results across various settings show SOTA performance in terms of accuracy and FID, with notable efficiency improvements compared to continuous methods. The method also integrates seamlessly with pre-trained text-to-image generation models, offering practical applications for controllable image generation.

**Strengths:**

1. Novelty: The approach is the first of its kind to apply compositional generation to discrete latent spaces, offering potential for increased efficiency and interpretability over traditional continuous sampling.
2. Applicability: Can be integrated with pretrained text-to-image models without fine-tuning, showing its broad applicability and ease of use.
3. Efficiency: The method is computationally efficient, delivering significant speed-ups over continuous methods, making it more practical for real-time or large-scale applications.

**Weaknesses:**

1. Although this work introduces the first approach for compositional image generation in discrete space, the methodology itself appears similar to ‘concept conjunction’ used in compositional diffusion models [1] within continuous space. Additionally, the rationale behind why the composition technique from continuous space models cannot be directly applied to discrete space remains unclear (as both approaches appear to rely on the composition of log-probabilities). Furthermore, the trade-offs mentioned in the introduction lack sufficient clarity and specificity.
2.	Limited Scope in Evaluated Datasets: The datasets are somewhat limited in complexity (CLEVR, FFHQ). More complex datasets with more varied attributes could better test the method’s generalization.

[1] Nan Liu, Shuang Li, Yilun Du, Antonio Torralba, and Joshua B Tenenbaum. Compositional visual generation with composable diffusion models. In Computer Vision–ECCV 2022: 17th European Conference, Tel Aviv, Israel, October 23–27, 2022, Proceedings, Part XVII, pages 423–439. Springer, 2022.

**Questions:**

1.	How does the proposed compositional VQ sampling method perform when applied to autoregressive-based models?
2.	Regarding the influence of weighting values on attributes:
 * Is the range of weights (e.g., -3 to +3) shown in Figure 6 applicable to all attributes, or does it require tuning for each attribute?
* When involving multiple components, how does varying the weight for each component affect the final generated result? Specifically, how do different weight values interact across components?

---

> ### Author Response · Authors · 2024-11-20
> **Response to reviewer**
>
> We thank the reviewer for their constructive comments, questions and suggestions. We respond below to each point under "weaknesses" and "questions" in the same order that they appear in the review.
>
> 1. Our derivation requires theoretical considerations above, beyond and independent to those of continuous processes, to which we dedicate over 2 full pages of our paper (spanning pages 4-6). We show empirically that the method does in fact work consistently with the theory, and not only this but it is dramatically faster (2.3-12 times) and dramatically more accurate (cutting error by 63%) than composed diffusion models. For instance, the elimination of the unconditional constant term (Appendix D.1) is crucial to our contribution (since there's no simple way to estimate the prevalence or density of a particular attribute in the data, especially for continuous variables) and is not a necessary consideration for continuous composition. To be more specific about the trade-offs offered by masked generative models (we also direct the reviewer to the cited literature, line 051): masked generative models allow trade-offs of sampling temperature (to control diversity vs. accuracy), the number of tokens added per timestep (to control quality vs. efficiency) and also the outright improvement of enabling generation of images larger than those in the training set in addition to in-painting and out-painting without any extra implementation.
> 2. Collectively, the 3 datasets studied represent very different compositional tasks. The two variants of CLEVR are actually very different from each other. Positional CLEVR requires the interpretation of a continuous input signal (object co-ordinates) while Relational CLEVR relies on the precise specification of attributes of pairs of objects as well as their relation to each other, making it far more difficult (reflected in the significantly larger error rate for all baseline methods compared to the other two datasets). The FFHQ dataset differs from the CLEVR datasets in a unique way: the there are only 3 binary input attributes (totalling 8 unique combinations), however realistic sampling requires far more in the way of diversity and coverage. We further demonstrated our method's generalisation to the text-to-image setting with a pre-trained model, without any fine-tuning or specialised training objective.
>
> Questions:
> 1. This is an interesting and insightful question, but unfortunately it lies beyond the scope of our present work with masked generative models. We suspect that in autoregressive models errors may accumulate more with time due to over-fitting spurious long-term correlations in the data, possibly resulting in poorer generations, however we leave implementation and extensive analysis and evaluation of compositional autoregressive models for image generation to future work.
> 2. Responding to each of the two sub-points:
>    * The range of weights does not require tuning: we used the same range $[-3,3]$ for generating a set of 7 images for the other two FFHQ attributes (see Appendix E.2). We used the value of w=3 for all quantitative experiments for all 3 datasets to achieve our SOTA results. We also trialled $w=2$ but found $w=3$ to give across-the-board accuracy improvements on all datasets, indicating that dataset-spcific tuning of $w$ is not necessary.
>    * The relative weight of each component specifies the relative importance of the corresponding feature being present in the generated image, at the cost of diversity. Specifying a higher weight for "glasses" (e.g. $w=3$) than for "smile" (e.g. $w=1$) means that the model will be more confident that the output image contains glasses, but the specific depiction of glasses will be less diverse (e.g. always sunglasses), meanwhile the same will not be true for smile (less confidence in the depiction of a smile but otherwise more diverse depictions of facial expression).

---

> ### Author Response · Authors · 2024-11-26
>
> Our earlier comment aimed to respond in detail to each of the reviewer's concerns and questions. We would like to check with the reviewer whether we have adequately addressed these concerns, before this stage of the discussion period ends?

---

> ### Author Response · Authors · 2024-11-28
>
> We would like to gently follow up from our previous message to check if the reviewer feels we have adequately addressed the concerns/questions, and if so if the reviewer might kindly increase their rating as a result?

---

### Official Review · Reviewer_qGWC · 2024-11-10

**Soundness:** 3
**Presentation:** 3
**Contribution:** 2
**Rating:** 5
**Confidence:** 4

**Summary:**

The paper introduces an composition algorithm that enables compositional generation (like score composition in EBM [1] or diffusion [2]) for discrete diffusion models or masked generation models. The method is straightforward, effective, and aligns well with intuitive understanding. The authors evaluate their approach on diverse datasets, including Positional CLEVR, Relational CLEVR, and FFHQ, demonstrating strong results.


[1] Du, Yilun, Shuang Li, and Igor Mordatch. "Compositional visual generation with energy based models." Advances in Neural Information Processing Systems 33 (2020): 6637-6647.

[2] Liu, Nan, Shuang Li, Yilun Du, Antonio Torralba, and Joshua B. Tenenbaum. "Compositional visual generation with composable diffusion models." In European Conference on Computer Vision, pp. 423-439. Cham: Springer Nature Switzerland, 2022.

**Strengths:**

- The study addresses a relatively unexplored area of compositional generation within discrete diffusion and masked generative models, complementing the majority of prior work focused on EBMs or continuous diffusion models. This contributes meaningfully to the field.
- The proposed approach is straightforward, effective, and intuitive, producing promising empirical results across various datasets.

**Weaknesses:**

- Lines 191-192 use the phrase “statistically independent of each other,” which may be ambiguous and potentially misleading. It could imply a graphical model with $c_i$ nodes pointing to $x$. My understanding is that equation (1) is derived directly from the product of experts assumption, similar to equation (4) in [1].
- The method essentially adopts a specialized form of classifier-free guidance (CFG), which has been explored in [3] and has been widely implemented in autoregressive and discrete diffusion models. This raises questions about the novelty of the approach.
- The paper provides results for compositional operations like AND and NOT, but does not address OR, leaving a gap in the evaluation.

[3] Nair, Nithin Gopalakrishnan, Wele Gedara Chaminda Bandara, and Vishal M. Patel. "Unite and conquer: Plug & play multi-modal synthesis using diffusion models." In Proceedings of the IEEE/CVF Conference on Computer Vision and Pattern Recognition, pp. 6070-6079. 2023.

**Questions:**

- It is unclear how equation (3) follows from equation (2). Could this approach face the same issues highlighted in [4]?

[4] Du, Yilun, Conor Durkan, Robin Strudel, Joshua B. Tenenbaum, Sander Dieleman, Rob Fergus, Jascha Sohl-Dickstein, Arnaud Doucet, and Will Sussman Grathwohl. "Reduce, reuse, recycle: Compositional generation with energy-based diffusion models and mcmc." In International conference on machine learning, pp. 8489-8510. PMLR, 2023.

---

> ### Author Response · Authors · 2024-11-20
> **Response to reviewer and details of revision**
>
> We thank the reviewer for their constructive comments, questions and suggestions. We address each of these points directly below and, where appropriate, the revised paper clarifies and addresses these points (specific line numbers below).
>
> Addressing the points under weaknesses in-order:
> 1. We thank the reviewer for drawing attention to lines 191 which previously had a typo. This line should state the assumption that the input conditions are independent of each other _conditional on x_, i.e. $P(c_1,c_2|x)\propto P(c_1|x)P(c_2|x)$ for a pair of attributes $(c_1,c_2)$, which the reviewer rightly states is the product of experts assumption. This is now clarified in the most recent revision (lines 194-195).
> 2. The reviewer's statement "essentially adopts ... classifier-free guidance (CFG)" is potentially misguided. The reviewer's reference [3] does not refer to classifier-free guidance nor to discrete generative models. The relationship between our method and masked CFG is analogous to the relationship between product of experts and a single classifier, i.e. they are conceptually related but otherwise distinct, with one being a prerequisite for the other. CFG does not offer any specific advantages for compositional image generation, which is the explicit focus of our contribution. Our motivation to implement and extensively evaluate our method (which is state-of-the art in accuracy and efficiency) is strongly informed by the significance of our result for composing the outputs of masked generative models, which requires theoretical considerations beyond those of continuous approaches (for instance the elimination of the conditioning terms from equation (2)).
> 3. This is not a gap in the evaluation; however, we accept that the reason for this should be made more explicit in the paper. In Boolean algebra, A OR B is identical to NOT (NOT A AND NOT B). This would correspond to $-(-a + -b)=a+b$ for conditional log-probabilities a and b, which is also the way we implement A AND B. This is a contradiction (A AND B is not equivalent to A OR B), which indicates that Boolean operations don't always translate to composition of log probability distributions: arithmetic negation is distributive, while Boolean negation is not, i.e. NOT(A AND B) does not imply NOT A AND NOT B. We have clarified this in Appendix D.4 of the most recent revision.
>
> In response to the reviewer's question under "questions":
> 1. We agree that the relationship between equations (2) and (3) is not immediately clear, primarily due to space limitations. We attempt to make this clearer below and this is now in Appendix D.2 of the latest revision (now referenced on line 223):
>
>    Equation (3) holds if and only if the joint distributions $s_t,c_i$ and $s_t,c_i$ are independent conditional on $s_{t+1}$, i.e. $P(s_t,c_i,c_j|s_{t+1})\propto P(s_t,c_i|s_{t+1})P(s_t,c_j|s_{t+1})$ for all pairs of distinct input conditions $c_i,c_j$ (this is the product of experts assumption, extended to sequential conditional generation).
>
>    $P(s_t,V|s_{t+1})=P(V|s_{t+1})$ for any random variable or collection of variables $V$, since $s_t$ is entirely determined by $s_{t+1}$ since the generation process is purely additive, meaning the representation of $s_t$ is contained by that of $s_{t+1}$. Therefore, the statement of the extended product-of-experts assumption can be simplified to:
>
>    $$P(c_i,c_j|s_{t+1})\propto P(c_i|s_{t+1})P(c_j|s_{t+1})$$
>
>    i.e. the extended product of experts assumption is equivalent to the original product of experts assumption and therefore (3) follows from (2) with the substitution of a single categorical variable $x$ with a stateful discrete generative process $s_{t+1}|s_t$, provided that the process is purely additive (such as unmasking or autoregressive sampling).

---

> > ### Comment · Reviewer_qGWC · 2024-11-25
> > **thanks for rebuttal**
> >
> > The reference [3] does say explicitly between Eq (9) and (10) their approach is a more general form of CFG, and Eq (10) is CFG or do I understand it wrongly?

---

> > > ### Author Response · Authors · 2024-11-26
> > > **Thanks for reply and request for further comments**
> > >
> > > Regarding the reviewer's clarification: We apologise for the inaccuracy in our initial rebuttal. The reviewer is correct: the generalisation of product of experts appears in [3]. [3] states clearly that this generalisation is for _Gaussian variables_ and is restricted to such. This form also appears multiple times throughout the literature and is not unique to [3] (it appears in [1] and [2]).
> > >
> > > However, our point regarding _discrete generative models_, and the relationship between CFG and our work, still stands. To our knowledge, there is presently nothing in the literature applying the compositional technique to discrete generative models. We believe that this is in part due to a previous lack of theoretical backing, which directly motivates our theoretical contribution, and the lack of empirical evidence that it works, which is another major part of our contribution.
> > >
> > > Our work therefore aims to directly addresses this gap in the literature: we offer the first derivation (to our knowledge) of the generalised product-of-experts for conditional discrete generative models, and we apply this in practice to achieve state-of-the-art accuracy and efficiency, with highly competitive FID scores, demonstrating a clear correspondence between theory and practice.
> > >
> > > We invite the reviewer to respond to the other 3 points in our rebuttal. We have responded fully to each of the reviewer's concerns, so we hope the reviewer will kindly comment on our other revisions and clarifications, especially in light of the presently borderline rating.

---

> ### Author Response · Authors · 2024-11-28
>
> We would like to gently follow up from our previous message to check if the reviewer feels we have adequately addressed the concerns/questions, and if so if the reviewer might kindly increase their rating as a result?

---

### Meta-Review · Area_Chair_5eky · 2024-12-20

**Metareview:**

The paper introduces an composition algorithm that enables compositional generation. It essentially adopts a specialized form of classifier-free guidance (CFG), which was well received. Two reviewers raised the concern over the novelty of this work, eps. given a relevant work titled "Unite and conquer: Plug & play multi-modal synthesis using diffusion models" and the rebuttal didn't address this concern. Besides, the experiments did not address some settings (e.g. OR) and were limited to some simple cases. Thus, I recommended a rejection.

**Additional Comments On Reviewer Discussion:**

The discussion during the rebuttal period did not lead to any changes.

---

### Decision · Program_Chairs · 2025-01-22

Reject